# Effect of Ultraviolet Irradiation on Vitamin D in Commonly Consumed Mushrooms in Thailand

**DOI:** 10.3390/foods12193632

**Published:** 2023-09-30

**Authors:** Kunchit Judprasong, Sochannet Chheng, Chanika Chimkerd, Sitima Jittinandana, Nattapol Tangsuphoom, Piyanut Sridonpai

**Affiliations:** 1Institute of Nutrition, Mahidol University, Phuttamonthon 4 Rd., Salaya, Phuttamonthon, Nakhon Pathom 73170, Thailand; kunchit.jud@mahidol.ac.th (K.J.); sochannetc@gmail.com (S.C.); sjittinandana@gmail.com (S.J.); nattapol.tng@mahidol.ac.th (N.T.); 2Department of Food Technology, Kampong Speu Institute of Technology, National 44 Rd. Thpong, Kampong Speu 050601, Cambodia; 3Center of Analysis for Product Quality, Natural Products Division, Faculty of Pharmacy, Mahidol University, 447 Sri-Ayuthaya Rd., Rajathevi, Bangkok 10400, Thailand; chanika.chm@mahidol.ac.th

**Keywords:** vitamin D, UV irradiation, LC-MS-MS, edible mushrooms, true retention

## Abstract

This study examined the effect and stability of ultraviolet B (UV-B) irradiation and subsequent cooking on vitamin D content in commonly consumed mushrooms in Thailand. Eight varieties of mushrooms were exposed to two-sided UV-B lamps for up to 3 h in a patented cabinet, followed by vitamin D content analysis. Thereafter, the four mushroom varieties with the highest vitamin D content were exposed to UV irradiation, cooked, and analyzed for various forms of vitamin D using LC-MS-MS. The results showed that vitamin D2 in all varieties of mushrooms significantly increased (*p* < 0.05) after UV-B irradiation according to the exposure time. The highest level of vitamin D2 was found in enokitake mushrooms. In addition, 25-OH D2 and vitamin D4 contents increased after UV-B irradiation in enokitake mushrooms. The vitamin D2 true retention in all cooked mushrooms ranged from 53 to 89% and was highest in stir-fried mushrooms. With economic investment, the two-sided UV-B cabinet has the potential to increase the vitamin D content in commercial mushroom production.

## 1. Introduction

Vitamin D is a calciferol and a fat-soluble vitamin comprising two core forms, namely, cholecalciferol (vitamin D3) and ergocalciferol (vitamin D2), which are structurally comparable secosteroids derived from the UV irradiation of provitamin D sterols [1]. A basic source of vitamin D3 in humans and many animals arises from the transformation of 7-dehydrocholesterol in the epidermis into vitamin D3 upon contact with ultraviolet (UV) irradiation in sunlight [2]. 

In human nutrition, vitamin D is essential for infants and young children to prevent rickets, while vitamin D deficiency in adults is related to osteoporosis. Vitamin D plays a crucial role in muscle growth and development and may help protect against neurodegenerative diseases, respiratory diseases in children, cardiovascular and autoimmune diseases, and both type 1 and type 2 diabetes, though current evidence for non-skeletal benefits is not yet conclusive [3,4]. Currently, very few diets contain naturally occurring vitamin D. Fish liver oils and the meat of fatty fish are among the greatest sources of vitamin D3. In Thailand, Tirakomonpong et al. (2019) [5] reported that Nile tilapia, common silver barb, and red Nile tilapia have high vitamin D content (19.8, 31.0, and 48.5 µg/100 g fresh weight (FW), respectively). More recently, Sridonpai et al. (2023) [6] reported that termite mushrooms and lung oyster mushrooms have high vitamin D content (7.15 ± 0.67 and 15.88 ± 7.31 µg/100 g edible portion, respectively).

In Thailand, vitamin D deficiency has been rising with prevalence rates for vitamin D insufficiency and vitamin D deficiency being reported as 45.2% (25(OH)D < 75 nmol/L) and 5.7% (25(OH)D < 50 nmol/L), respectively [7]. Consequently, vitamin D deficiency is now an important public health issue that must be addressed through greater efforts to increase dietary intake that takes into account changing lifestyles and the prevailing environment. Moreover, fortification of vitamin D in suitable food products marketed in Thailand is not yet mandatory [8]. Research on high vitamin D content in Thailand can inform public health policies and guidelines, including recommendations for supplementation, fortification of foods, and public-awareness campaigns to promote the health benefits of vitamin D.

Many studies have reported that mushrooms are respectable dietary sources of vitamin D, particularly vitamin D2 [3] and ergosterol (a precursor of vitamin D2), though in varying amounts [9] depending especially on growing conditions. For instance, fresh white button mushrooms have a vitamin D2 content of less than 0.5 µg/ 100 g or 20 IU since they are grown in dark rooms and not exposed to sunlight or ultraviolet (UV) radiation [10]. 

An effective method to increase the amount of vitamin D2 is to expose mushrooms to controlled and specific amounts of UV radiation using a fluorescent or pulsed lamp. The greatest effective wavelength to generate vitamin D2 production in mushrooms is UV-B irradiation (280–315 nm) [11]. Several studies have confirmed that when fresh mushrooms are intentionally exposed to post-harvest UV-B irradiation, they generate significant amounts of vitamin D2 [12,13,14,15]. Moreover, an increase in vitamin D4 content was found in oyster mushrooms from 0 to 20 µg/g of dried matter when exposed to UV-B lamp radiation for only 30 min [3]. Consequently, UV-B-irradiated mushrooms can be important dietary sources of vitamin D2 [13,16].

Thai people consume different varieties of mushrooms that are normally prepared by boiling or stir-frying, based on the Food Consumption Survey in Thailand (2015) [17]. Unfortunately, there is a scarcity of data on the effects of UV irradiation and cooking after UV irradiation on commonly consumed mushrooms in Thailand. Consequently, this study aimed to assess the effect of UV-B irradiation on eight varieties of commonly consumed mushrooms in order to identify the optimum conditions for UV-B irradiation. Thereafter, the four varieties of mushrooms with the greatest increase in vitamin D content were evaluated in terms of the effect of cooking (boiling, stir-frying, and grilling) after ultraviolet irradiation on vitamin D content. An assessment of the post-ultraviolet irradiation stability of vitamin D was also conducted.

## 2. Materials and Methods

### 2.1. Sample Preparation

Eight types of commonly consumed fresh mushrooms were purchased from a wholesale market (Fai Fah market, Nakhon Chai Si district, Nakhon Pathom province, Thailand) in the morning (approximately 8 a.m.) and then placed in an icebox and directly transported to the laboratory at the Institute of Nutrition, Mahidol University (INMU) (Table 1). To assess the effect of irradiation, the mushrooms were trimmed to remove inedible portions if needed. For each mushroom type, the edible portion was weighed and sized into large, medium, and small amounts. These samples (2 kg) were then divided into six groups of the same amount (about 300 g/group) and each group containing large, medium, and small sizes before being placed into a UV cabinet to be exposed to UV-B irradiation for 0, 0.25, 0.5, 1, 2, and 3 h (six study groups). The distance from the UV lamp (18 W UV-B daylight reptile fluorescent lamp) to each sample was set at 15 cm. The UV cabinet was registered and accepted as a petty patent by the Department of Intellectual Property, Thailand, under the name “UV-B cabinet” with a two-sided light source to increase vitamin D2 and D4 contents (Figure 1) [18]. The UV-B treated samples were homogenized in a food blender (MR-1268, MARA HOUSE Co., Ltd., Bangkok, Thailand. For freeze-drying, homogenized UV-B treated samples were put in freeze-dryer trays and kept in a freezer at −20 °C for more than 4 h prior to their being transferred to the freeze-dryer machine. Freeze-dried samples were dried in the freeze-dryer system (Heto Powerdry PL 9000 Freeze Dryer, Corston, UK) until completely dried (2–3 days). After that, the samples were ground into homogenized samples (MR-1268, MARA HOUSE Co., Ltd., Bangkok, Thailand) and kept at −20 °C until vitamin D analysis. 

### 2.2. Vitamin D Content after Exposure to UV Irradiation

#### 2.2.1. Mushroom Color Determination

The mushroom color was determined by comparing mushroom samples before and after UV-B irradiation and at different times to the Munsell color system (Munsell^®^, Baltimore, MD, USA) and reported as the Munsell color value.

#### 2.2.2. Vitamin D Content Determination

Determination of vitamin D was performed in duplicate using the method described by Suthipibul et al. (2020) [19]. First, 4 ppm of trideuterated ^2^H_3_D2 (50 µL) was used as an internal standard. Samples were saponified using methanol (6 mL) and 45% potassium hydroxide (4 mL), and sodium ascorbate was used as an antioxidant. Acetonitrile (5 mL) and ether:pentane (20:80) (20 mL) were used as extraction solvents, while 10% NaCl (10 mL) and ether:pentane (20:80) (10 mL) were used for repeat extraction. Methanol (1.9 mL) and DI water (0.1 mL) were used for reconstitution. A 0.2 µm NYLON syringe filter (Filtrex Inc., Hillview Avenue, Singapore) was used as a filter before injection into the UHPLC-MS-MS system.

A UHPLC (ultimate 3000 system, ThermoFisher, Waltham, MA, USA), connected with a reversed-phase column (Water HSS T3 2.1 × 150 mm, and 1.8 µm particle size) (Acquity uplc, Waters Corporation, Milford, MA, USA), was used to separate the various forms of vitamin D. The mobile phase comprised 2 mM ammonium formate in methanol and 2 mM ammonium formate in water. The flow rate of the mobile phase was 500 µL/min, the temperature of the column was set at 60 °C, and the injection volume was 20 µL. TSQ Quantis tandem mass spectrometry (ThermoFisher, Waltham, MA, USA) was used for quantitative measurement. Atmospheric pressure chemical ionization (APCI) was used as the ionization method for mass spectrometry. Quantitative analysis was performed by MS-MS mode for various forms of vitamin D. The following MS-MS parameters were used: 350 °C of vaporizer temperature, 325 °C of iron transfer tube temperature, 45 Arbitrary units (Arb) of sheath gas, 5 Arb of auxiliary gas, 1 Arb of sweep gas, and 6.5 µA of a positive ion discharge current. Seven vitamers of vitamin D (Vitamin D2, Vitamin D3, Ergosterol, 7-DHC, 25-OH D2, 25-OH D3, and Vitamin D4) and ^2^H^3^D_2_ (internal standard) were separated and detected by UHPLC-MS-MS). As an example, the chromatograms of the vitamin D components are shown in Figure 2.

### 2.3. Effect of Cooking on High-Vitamin-D Mushrooms

Four mushroom varieties having the uppermost vitamin D content after UV-B irradiation under the optimum conditions were selected to study the effect of cooking. Samples were cooked by stir-frying (with a small amount of palm oil), boiling, and grilling according to common household cooking methods (Table 2). Yield factor, weight loss, and moisture contents of cooked mushrooms were assessed. Raw and cooked mushrooms were blended in a food blender. Thereafter, the samples were prepared for freeze-drying, ground into fine powder, and kept at −20 °C until vitamin D analysis.

#### 2.3.1. Yield Factor Determination

Yield factor data were calculated from the weight of edible portion before and after cooking according to the formula below.
$$\%\;\mathrm{Yield}\;\mathrm{factor}\;=\frac{\mathrm{Weight}\;\mathrm{of}\;\mathrm{edible}\;\mathrm{portion}\;\mathrm{of}\;\mathrm{cooked}\;\mathrm{sample}\;\left(g\right)}{\mathrm{Weight}\;\mathrm{of}\;\mathrm{edible}\;\mathrm{portion}\;\mathrm{of}\;\mathrm{raw}\;\mathrm{sample}\;\left(g\right)}\times 100$$

#### 2.3.2. Weight Loss Determination

Weight loss data were calculated from the weight of edible portion before and after cooking according to the formula below.
$$\%\;\mathrm{Weight}\;\mathrm{loss}\;=\frac{\mathrm{Weight}\;\mathrm{of}\;\mathrm{raw}\;\mathrm{sample}\;\left(g\right)-\;\mathrm{weight}\;\mathrm{of}\;\mathrm{cooked}\;\mathrm{sample}\;\left(g\right)}{\mathrm{Weight}\;\mathrm{of}\;\mathrm{edible}\;\mathrm{portion}\;\mathrm{of}\;\mathrm{raw}\;\mathrm{sample}\;\left(g\right)}\times 100$$

#### 2.3.3. True Retention of Vitamin D

Samples were weighed (to at least 2 significant digits) before and after cooking. The data gained, combined with the amounts of vitamin D in raw and cooked mushrooms, were applied to calculate true retention [20]. 

#### 2.3.4. Moisture Determination

The moisture contents of fresh samples were determined in duplicate according to the Association of Official Analytical Chemists (AOAC) method no. 925.45, 2019 [21]. Moisture content was calculated based on weight loss after drying at 100 + 1 °C.

### 2.4. Statistical Analysis

The vitamin D content of each mushroom sample was presented as mean ± SD. Analysis of variance (two-way ANOVA) and Duncan’s multiple range test were used to compare mushroom varieties by UV time and cooking methods to determine the significance of differences (*p* ≤ 0.05). Correlation analysis was used to identify the relationship of increasing change between vitamin D2 and vitamin D4 contents in mushrooms after exposure to UV-B irradiation. Statistical analysis was performed by using SPSS Statistics for Windows, Version 19.0.

## 3. Results and Discussion

### 3.1. Physical Characteristics of Raw Mushroom Samples

Most mushrooms of the same species including oyster mushrooms (Bhutan and lung oyster mushrooms) and shimeji mushrooms have similar characteristics (size of the cap, stalk, and height). However, they are visually different in color, which is a factor in identifying and confirming different varieties of mushrooms even in the same species (e.g., white shimeji and brown hon shimeji mushrooms) (Table 3 and Table 4). 

### 3.2. Effect of Ultraviolet (UV) Irradiation on Vitamin D Contents in Mushrooms

#### 3.2.1. Vitamin D2 Content

The vitamin D2 contents of the different varieties of mushrooms exposed to UV-B irradiation at different times are shown in Table 5, Table 6, Table 7 and Table 8. The control mushroom (unexposed to UV-B) that showed the highest vitamin D2 content was the white shimeji mushroom, 83.73 ± 12.79 µg/100 g dried weight (DW) (7.96 ± 1.22 µg/100 g fresh weight, FW), while the lowest contents were in wood ear and straw mushrooms, which were undetectable and found to be less than 1 µg/100 g DW in enokitake, shiitake, and brown hon shemeji mushrooms. Most vitamin D2 content levels of mushrooms in this study were found in the same range of raw cultivated mushrooms purchased from retail shops sold in the UK, Europe, North America, Australia, and New Zealand, which commonly reported less than 1 µg/100 g FW [2,12,13,22,23].

After UV-B irradiation, the overall amount of vitamin D2 in mushrooms increased significantly (*p* < 0.05) after exposure, which agrees with a previous study that reported that vitamin D2 content in mushrooms increased after UV exposure [16]. In addition, Keegan et al. (2013) [24] reported that mushrooms exposed to UV-B radiation contain a significant amount of vitamin D2. For the enokitake mushroom, the amount of vitamin D2 increased significantly (*p* < 0.05) to the highest amount, from 6.68 ± 1.42 to 1880.22 ± 197.76 µg/100 g DW (0.77 ± 0.16 to 271.59 ± 28.57 µg/100 g FW), after UV-B irradiation from 0 to 3 h, followed by Bhutan oyster, lung oyster, and brown hon shimeji mushrooms. While not being detected for both wood ear and straw mushrooms before UV-B irradiation, the amount of vitamin D2 in these mushrooms increased significantly (*p* < 0.05) after exposure to irradiation for 3 h. This finding is similar to a study by Hu et al. (2020) [24], which reported that the amount of vitamin D2 increased from an undetectable amount to 23.71 ± 5.72 μg/g DW in dry oyster mushroom powder upon UV irradiation.

The mushrooms with the highest levels of vitamin D2 were the enokitake mushroom, brown hon shimeji mushroom, Bhutan oyster mushroom, and lung oyster mushroom. Consequently, these varieties were selected for studying the effect of cooking on vitamin D retention. In addition, the optimum condition for UV-B irradiation (one hour at a distance of 15 cm from the UV-B lamp) was identified based on three main factors: color change, increasing vitamin D2 content, and the price of the mushrooms. 

#### 3.2.2. Ergosterol Content

High amounts of ergosterol in the cultivated mushrooms were found (Table 5, Table 6, Table 7 and Table 8). The ergosterol contents in several varieties of cultivated mushrooms were not significantly different (*p* < 0.05) before and after UV-B irradiation including wood ear, Bhutan oyster, white shimeji, brown hon shimeji, and lung oyster mushrooms. For the enokitake mushroom, the ergosterol content appeared to decrease by 19% from 140.58 ± 3.97 to 114.29 ± 11.92 mg/100 g DW (16.30 ± 0.46 to 14.29 ± 1.49 mg/100 g FW) after exposure to UV-B for 1 h. The amount of ergosterol in the lung oyster mushroom decreased slightly when exposed to UV-B for 3 h. Further, the level of ergosterol in the shiitake mushroom decreased by 40% from 211.48 ± 9.91 to 137.16 ± 18.00 mg/100 g DW (34.91 ± 1.64 to 27.58 ± 3.62 mg/100 g FW) after exposure to UV-B for 3 h. This finding agreed well with the results of the study of Hu et al. (2020) [25], which reported that the amount of ergosterol decreased with an increase in vitamin D2, whereas most of the ergosterol was probably UV-degraded. Jasinghe and Perera (2005) [16] and Perera (2003) [26] reported that, in shiitake mushrooms (fresh), the amount of ergosterol was highest in the gills, followed by the cap and stalk, with the gills having twice the amount of ergosterol compared to the cap. Moreover, the amounts of vitamin D2 in common mushroom varieties generally depend on several factors, such as time of day, season, latitude, weather conditions, surface area, exposure time, and the morphology of each mushroom variety.

#### 3.2.3. The 25-Hydroxyl Vitamin D Content

The content of 25-OH D2 in both control (unexposed) and treated samples (exposed) was found to be highest (significance: *p* < 0.05) in white shimeji mushrooms, followed by wood ear, straw, and enokitake mushrooms (Table 5, Table 6, Table 7 and Table 8). The amount of 25-OH D2 in white shimeji mushrooms increased by 43% to the highest amount, from 4145.52 ± 694.53 to 5941.70 ± 991.57 µg/100 g DW (394.00 ± 66.01 to 589.05 ± 98.30 µg/100 g FW), when exposed to UV-B irradiation for 15 min. In both wood ear mushroom and enokitake mushroom, the amounts of 25-OH D2 increased by 24% (from 515.14 ± 7.26 to 666.29 ± 92.98 µg/100 g DW) and 109% (from 44.96 ± 10.23 to 93.98 ± 2.99 µg/100 g DW) when exposed to UV-B for 0–3 h, respectively. Consumption of UV-B-irradiated mushrooms may increase serum 25(OH)D when baseline vitamin D status is low via an increase in 25(OH)D2 (24.2 nmol/L) [27]. Urbain (2011) [28] also investigated the bioavailability of vitamin D2 from UV-B-irradiated button mushrooms and found a high level of 25-OH D2 (3.8 to 5.7 nmol/L) in human serum from enhanced mushrooms exposed to UV-B.

#### 3.2.4. Vitamin D4 Content

A study by Phillips et al. (2012) [2] confirms recent research findings that have identified vitamin D4 and provitamin D4 in various edible mushroom species. In the present study, vitamin D4 content in the cultivated mushrooms varied in both the control (unexposed UV-B) and the UV-B exposed samples (Table 5, Table 6, Table 7 and Table 8). After irradiation, the amount of vitamin D4 in all types of mushrooms increased according to the time of UV-B exposure from 0.25 to 3 h. The highest concentration of vitamin D4 was found in the enokitake mushroom, which increased (2301%) significantly (*p* < 0.05), from 282.59 ± 0.34 to 6504.61 ± 21.74 µg/100 g DW, which was 29 times higher after being exposed to UV-B from 0 to 3 h, followed by lung oyster and Bhutan oyster mushroom. In addition, vitamin D4 content in brown hon shimeji mushrooms increased (552%) significantly (*p* < 0.05), from 261.23 ± 10.60 to 1702.47 ± 0.93 µg/100 g DW. A study by Krings and Berger (2014) [29] reported that UV-B lamp irradiation led to an increase in vitamin D4 content in oyster mushrooms from 0 to 20 μg/g DW after only 30 min of exposure. The 22,23-dihydroergosterol (provitamin D4) in mushrooms is converted to vitamin D4. All commonly consumed mushrooms contain provitamin D4, making them a probable source of vitamin D4 if exposed to UV radiation.

#### 3.2.5. The Correlation between Vitamin D2 and Vitamin D4 Contents in Mushrooms after Exposure to UV-B Irradiation

The same trend of increasing vitamin D2 and vitamin D4 contents when exposed to UV-B irradiation for 0–3 h is shown in Table 5, Table 6, Table 7 and Table 8. Statistical analysis of the correlation, vitamin D2, and vitamin D4 contents showed a statistically significant linear relationship (r = 0.893, *n* = 96, *p* < 0.01). The direction of the relationship is positive, meaning that vitamin D2 and vitamin D4 in all types of mushrooms tend to increase together after exposure to UV irradiation for 0–3 h. This finding agrees well with those of previous studies on vitamin D4 in mushrooms after exposure to UV-B irradiation that showed a positive correlation between vitamin D2 and vitamin D4 [2,30].

### 3.3. Effect of Cooking in Selected Mushrooms

#### 3.3.1. Edible Portion

Mushrooms tend to contain a high percentage of edible portion (over 80%). The percentages of edible portions in the cultivated mushrooms differed depending on variations within mushroom species (Table 9). The results showed that the percentages of edible portion for the cultivated mushrooms were within the same range of 80–95%. Hence, whole mushrooms were prepared as an edible portion and used for moisture and vitamin D analyses.

#### 3.3.2. Yield Factor

Boiled and stir-fried cultivated mushrooms retained yield factors ranging from 66–88% to 69–83% (Table 9). However, the grilling method used for the four varieties of cultivated mushrooms led to lower yield factors than the other cooking methods (60–64%). The cooking yield in this study agrees well with that reported in a study by Ložnjak and Jakobsen (2018) [31]. Hence, different yield factors of cultivated mushrooms could be affected by different cooking methods.

#### 3.3.3. Moisture Content

The moisture content of the cultivated mushrooms ranged from 88 to 92 g per 100 g edible portion (EP) (Table 9). Similar moisture content levels were found in the United States Department of Agriculture Food Composition Database, 2021 [32], as well as in a study by Kumar (2013) [33], which reported that fresh mushrooms contained about 85–95% moisture. The moisture contents for boiled samples were slightly higher (ranging between 93 and 94%) than for fresh samples, while the stir-fried and grilled samples had lower moisture content compared to uncooked samples. The value of moisture content in the stir-fried sample was the lowest, ranging from 80 to 85%. Similar results were observed in a study by Roncero-Ramos et al. (2017) [34], where the moisture content in fried mushrooms was found to be lowest compared with other cooking methods, such as boiling, microwaving, and grilling.

#### 3.3.4. Effect of Different Cooking Methods on Vitamin D Content in Mushrooms

The true retention of vitamin D2 in the enokitake mushroom was highest for the stir-fried method (89.3 ± 9.8%), followed by boiling (73.2 ± 6.7%), while the lowest was for grilling (68.2 ± 13.4%) (Table 10). However, there were no significant differences (*p* < 0.05) among these three cooking methods. The true retention of vitamin D2 in all mushrooms ranged from 53 to 89%, which is similar to findings in a study by Ložnjak and Jakobsen (2018) [31], where the true retention of vitamin D2 in mushrooms was significantly lower than 100%, with retention rates ranging from 62 to 88%. Among these three cooking methods, the true retention of vitamin D2 was highest in the Bhutan oyster mushroom compared to the other three mushrooms in terms of boiling (88.8 ± 9.7%), stir-frying (87.4 ± 14.6%), and grilling (87.2 ± 19.9%). Loss of vitamin D2 during household cooking could be attributed to high temperatures for all cooking methods and some partial loss due to leakage into the cooking oil during stir-frying. 

Overall, we found that the true retention of vitamin D4 was similar in trend to that of the true retention of vitamin D2. However, while the percentage of true retention for vitamin D4 in cooked mushrooms has not been researched thoroughly, studies exist on vitamin D4 in mushrooms after exposure to UV-B irradiation that found a positive correlation between vitamin D4 and vitamin D2 [2,30]. This study also found a positive correlation between vitamin D2 and vitamin D4 contents in mushrooms after exposure to UV-B irradiation. In contrast, the results of true retention of 25-OH D2 in all mushroom varieties were highest for grilling. The true retention of 25-OH D2 among the three cooking methods ranged from 46 to 100%. The retention of 25-OH D2 among these four mushrooms was highest in enokitake mushrooms, followed by Bhutan oyster, lung oyster, and brown hon shimeji mushrooms.

## 4. Conclusions

UV-B irradiation is an interesting way to increase vitamin D content in mushrooms. One hour of UV-B irradiation at a distance of 15 cm from the UV-B lamp is the optimum condition identified in this study. The amount of vitamin D2 and vitamin D4 in all mushrooms increased significantly (*p* < 0.05) after irradiation. Cooking may cause a loss of vitamin D content, but the degree of loss depends on the cooking time, heating process, and physical characteristics of the mushrooms. Consuming one portion (40 g) of cooked mushrooms after exposure to UV irradiation provides an excellent source of vitamin D. However, this study also has limitations in terms of sample collection and different sizes of mushrooms, which may affect the increasing level of vitamin D.

## Figures and Tables

**Figure 1 foods-12-03632-f001:**
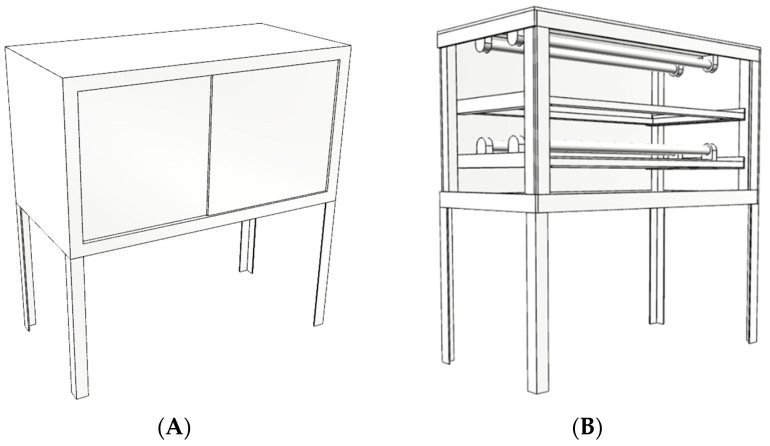
Picture of the UV cabinet. (**A**) Exterior of the UV cabinet. (**B**) Interior of the UV cabinet.

**Figure 2 foods-12-03632-f002:**
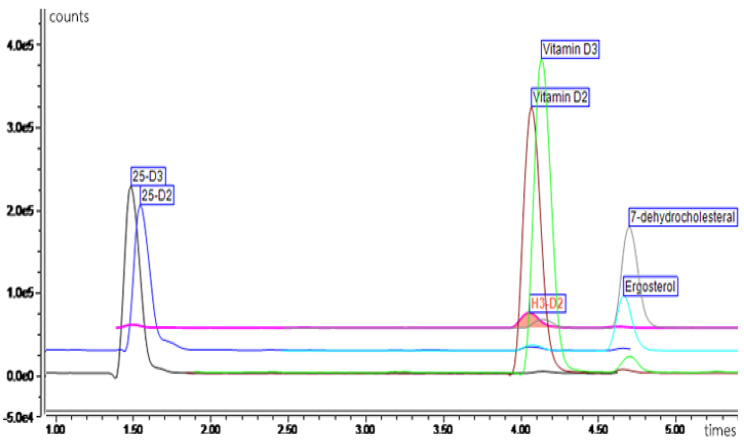
UHPLC-MS-MS chromatograms of vitamin D components.

**Table 1 foods-12-03632-t001:** List of selected cultivated mushrooms.

English Name	Thai Name	Scientific Name
Bhutan oyster mushroom	Het-Bhutan	*Pleurotus sajor-caju (Fr.) Singer*
Brown hon shimeji mushroom	Het-Lin-Dam	*Hypsizygus tessellatus*
Enokitake mushroom	Het-Khem-Thong	*Flammulina velutipes*
Lung oyster mushroom	Het-Nang-Fa	*Pleurotus pulmonarius*
Shiitake mushroom	Het-Hom	*Lentinula edodes*
Straw mushroom	Het-Fang	*Volvariella volvacea*
White shimeji mushroom	Het-Lin-Khao	*Hypsizygus tessellatus*
Wood ear mushroom	Het-Hu-Nu	*Auricularia auricula-judae*

**Table 2 foods-12-03632-t002:** Common household methods and conditions for cooking of high-vitamin-D mushrooms.

Mushroom	Cooking Time (min)		
	Boiling	Stir-Frying	Grilling
	(85–100 °C)	(110–150 °C)	(150–200 °C)
Bhutan oyster mushroom (*Pleurotus sajor-caju (Fr.) Singer*)	10	4	6
Brown hon shimeji mushroom (*Hypsizygus tessellatus*)	5	4	6
Enokitake mushroom (*Flammulina velutipes*)	4	3	5
Lung oyster mushroom (*Pleurotus pulmonarius*)	10	4	6

**Table 3 foods-12-03632-t003:** Appearance and physical characteristics of five cultivated mushrooms.

Type	Appearance	Cap Size Average (cm) (Range)	Stalk Size Average (cm) (Range)	Height Average (cm) (Range)
Bhutan oyster mushroom *(Pleurotus sajor-caju (Fr.) Singer)*	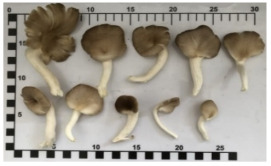	7.27.2 (2.4–12)	5 (4–6)	8.8 (4.7–12.9)
Brown hon shimeji mushroom *(Hypsizygus tessellatus)*	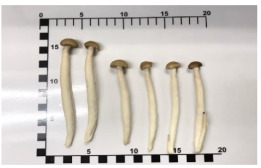	2.0 (1–3)	12 (9–14)	12 (10–14)
Enokitake mushroom *(Flammulina velutipes)*	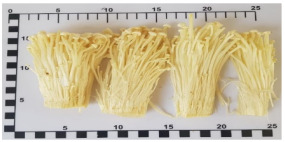	0.4 (0.2–0.6)	13 (12–14)	12 (10–14)
Lung oyster mushroom *(Pleurotus pulmonarius)*	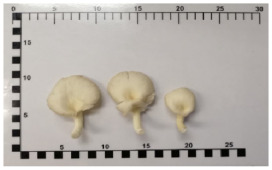	7.5 (4–11)	6 (4–8)	11.2 (7–15.5)
Shiitake mushroom *(Lentinula edodes)*	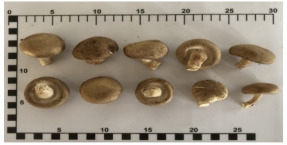	5.2 (4.5–5.8)	3 (3–4)	4.4 (3.8–5)

**Table 4 foods-12-03632-t004:** Appearance and physical characteristics of three cultivated mushrooms.

Type	Appearance	Cap Size Average (cm) (Range)	Stalk Size Average (cm) (Range)	Height Average (cm) (Range)
Straw mushroom *(Volvariella volvacea)*	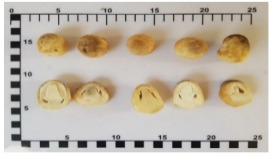	3.5 (2.3–4.7)	N/A	4.1 (2.5–5.7)
White shimeji mushroom *(Hypsizygus tessellatus)*	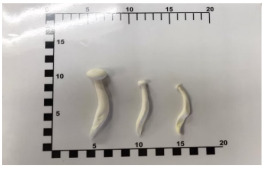	2 (0.5–3)	11 (9–12)	11 (10–12)
Wood ear mushroom *(Auricularia auricula-judae)*	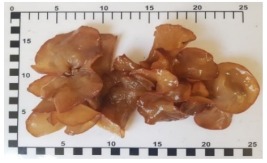	5.8 (3–8.9)	1.5 (1–2)	7 (3.2–10.7)

N/A = Not applicable due to lack of stalk.

**Table 5 foods-12-03632-t005:** Vitamin D contents and color change in Bhutan oyster and brown hon shimeji mushrooms after exposure to UV irradiation for 0–3 h.

Mushrooms	UV Time (Hour)	Vitamin D2 (µg/100 g DW)	25-OH D2 (µg/100 g DW)	Ergosterol (mg/100 g DW)	Vitamin D4 (µg/100 g DW)	Color (Munsell Code)
Bhutan oyster mushroom (*Pleurotus sajor-caju (Fr.) Singer*)	0	22.95 ± 0.30 ^f^	1925.52 ± 210.89 ^a^	114.92 ± 4.91 ^a^	240.46 ± 20.43 ^e^	C: 10YR 4-6/2 S: 10YR 8-9/2
	0.25	291.31 ± 19.62 ^e^	1514.86 ± 37.12 ^a^	125.08 ± 8.96 ^a^	706.59 ± 7.52 ^d^	C: 10YR 4-6/2 S: 10YR 8-9/2
	0.5	530.23 ± 19.96 ^d^	2040.66 ± 240.66 ^a^	122.81 ± 4.00 ^a^	960.66 ± 6.24 ^c^	C: 10YR 4-6/2 S: 10YR 8-9/2
	1	920.19 ± 8.71 ^c^	1869.49 ± 4.59 ^a^	99.25 ± 4.06 ^a^	1640.11 ± 67.36 ^b^	C: 10YR 4-6/2 S: 10YR 8-9/2
	2	1052.16 ± 72.65 ^b^	1550.32 ± 231.30 ^a^	96.60 ± 4.84 ^a^	1587.08 ± 10.21 ^b^	C: 10YR 4-6/2 S: 10YR 7-9/2
	3	1468.83 ± 34.96 ^a^	1574.76 ± 11.69 ^a^	99.22 ± 11.40 ^a^	2550.12 ± 106.25 ^a^	C: 10YR 4-5/2 S: 10YR 8/2
Brown hon shimeji mushroom (*Hypsizygus tessellatus*)	0	9.00 ± 1.58 ^f^	2392.79 ± 41.73 ^a^	138.12 ± 6.82 ^a^	261.23 ± 10.60 ^f^	C: 10YR: 6-7/2 S: 5YR: 9/1
	0.25	149.94 ± 2.82 ^e^	3020.79 ± 398.41 ^a^	127.14 ± 15.10 ^a^	478.54 ± 12.42 ^e^	C: 10YR: 6-7/2 S: 5YR: 9/1
	0.5	301.80 ± 5.76 ^d^	2368.39 ± 69.06 ^a^	148.83 ± 3.64 ^a^	742.83 ± 11.18 ^d^	C: 10YR: 6-7/2 S: 5YR: 9/1
	1	624.83 ± 30.14 ^c^	2240.45 ± 363.57 ^a^	124.26 ± 13.36 ^a^	1029.53 ± 65.52 ^c^	C: 10YR: 6-7/2-4 S: 5YR: 9/1
	2	935.40 ± 13.99 ^b^	2035.30 ± 327.08 ^a^	122.70 ± 7.76 ^a^	1431.03 ± 55.06 ^b^	C: 10YR: 6-7/2 S: 5YR: 9/1
	3	1030.29 ± 19.58 ^a^	2476.84 ± 218.03 ^a^	102.40 ± 21.02 ^a^	1702.47 ± 0.93 ^a^	C: 10YR: 6-7/2-4 S: 5YR: 9/1

Note: C refers to the mushroom cap, while S refers to the mushroom stalk/stem. Different letters in the same column indicate significant differences (*p* < 0.05).

**Table 6 foods-12-03632-t006:** Vitamin D contents and color change in enokitake and lung oyster mushrooms after exposure to UV irradiation for 0–3 h (cont.).

Mushrooms	UV Time (Hour)	Vitamin D2 (µg/100 g DW)	25-OH D2 (µg/100 g DW)	Ergosterol (mg/100 g DW)	Vitamin D4 (µg/100 g DW)	Color (Munsell Code)
Enokitake mushroom (*Flammulina velutipes*)	0	6.68 ± 1.42 ^d^	44.96 ± 10.23 ^bc^	140.58 ± 3.97 ^abc^	282.59 ± 0.34 ^f^	C: 5Y 9/2 S: 5Y 9/2
	0.25	414.01 ± 20.37 ^cd^	69.44 ± 7.72 ^bc^	130.66 ± 24.28 ^bc^	1710.67 ± 109.76 ^e^	C: 5Y 9/2 S: 5Y 9/2
	0.5	615.18 ± 22.55 ^c^	30.13 ± 1.05 ^c^	110.35 ± 18.12 ^c^	2777.47 ± 162.21 ^d^	C: 5Y 9/2 S: 5Y 9/2
	1	1226.59 ± 100.69 ^b^	40.67 ± 6.81 ^bc^	114.29 ± 11.92 ^bc^	4282.19 ± 3.16 ^c^	C: 5Y 9/2-3 S: 5Y 9/2-3
	2	1349.06 ± 59.34 ^b^	73.61 ± 22.81 ^ab^	152.63 ± 4.59 ^ab^	4973.55 ± 6.05 ^b^	C: 5Y 9/3 S: 5Y 9/3
	3	1880.22 ± 197.76 ^a^	93.98 ± 2.99 ^a^	151.93 ± 10.80 ^a^	6504.61 ± 21.74 ^a^	C: 5Y 9/3 S: 5Y 9/3
Lung oyster mushroom (*Pleurotus pulmonarius*)	0	72.40 ± 0.20 ^e^	108.11 ± 4.32 ^a^	229.10 ± 53.57 ^a^	509.62 ± 34.12 ^d^	C: 5Y 8.5-9/2 S: 5Y 8.5-9/2
	0.25	258.95 ± 22.88 ^de^	85.06 ± 22.24 ^a^	222.12 ± 20.49 ^a^	860.09 ± 21.42 ^d^	C: 5Y 8.5-9/2 S: 5Y 8.5-9/2
	0.5	457.47 ± 29.44 ^d^	124.89 ± 16.95 ^a^	237.97 ± 9.99 ^a^	1350.72 ± 33.33 ^c^	C: 5Y 8-9/2 S: 5Y 8-9/2
	1	704.73 ± 4.48 ^c^	122.93 ± 21.86 ^a^	232.49 ± 18.22 ^a^	2143.33 ± 160.58 ^b^	C: 5Y 8-9/2 S: 5Y 8-9/2
	2	1063.54 ± 101.83 ^b^	103.93 ± 1.70 ^a^	210.74 ± 17.95 ^a^	2200.19 ± 151.90 ^b^	C: 5Y 8-8.5/2 S: 5Y 8-8.5/2
	3	1519.13 ±114.79 ^a^	104.28 ± 38.79 ^a^	167.33 ± 6.29 ^a^	2731.37 ± 140.46 ^a^	C: 5Y 8-8.5/2 S: 5Y 8-8.5/2

Note: C refers to the mushroom cap, while S refers to the mushroom stalk/stem. Different letters in the same column indicate significant differences (*p* < 0.05).

**Table 7 foods-12-03632-t007:** Vitamin D contents and color change in shiitake and straw mushrooms after exposure to UV irradiation for 0–3 h (cont.).

Mushrooms	UV Time (Hour)	Vitamin D2 (µg/100 g DW)	25-OH D2 (µg/100 g DW)	Ergosterol (mg/100 g DW)	Vitamin D4 (µg/100 g DW)	Color (Munsell Code)
Shiitake mushroom (*Lentinula edodes*)	0	5.93 ± 2.29 ^f^	522.23 ± 11.05 ^a^	211.48 ± 9.91 ^a^	381.83 ± 5.49 ^a^	C: 10YR 6-7/4 S: 10YR 9/1-2
	0.25	46.99 ± 3.72 ^e^	609.21 ± 42.16 ^a^	178.25 ± 13.98 ^abc^	378.13 ± 55.02 ^a^	C: 10YR 5-6/4 S: 10YR 9/1-2
	0.5	91.06 ± 6.71 ^d^	623.52 ± 105.09 ^a^	181.79 ± 8.77 ^abc^	399.71 ± 34.17 ^a^	C: 10YR 4-6/4 S: 10YR 9/1-2
	1	142.42 ± 3.10 ^c^	641.77 ± 39.43 ^a^	199.55 ± 1.28 ^bc^	463.18 ± 17.51 ^a^	C: 10YR 4-6/4 S: 10YR 8-9/2
	2	193.14 ± 11.87 ^b^	624.69 ± 35.53 ^a^	145.52 ± 7.68 ^c^	364.20 ± 1.34 ^a^	C: 10YR 4-6/4 S: 10YR 8-9/1-2
	3	270.44 ± 2.77 ^a^	533.65 ± 114.36 ^a^	137.16 ± 18.00 ^bc^	413.64 ± 31.08 ^a^	C: 10YR 3-5/4 S: 10YR 8-9/1-2
Straw mushroom (*Volvariella volvacea*)	0	0.00 ^f^	128.20 ± 5.27 ^b^	2.59 ± 0.21 ^b^	0.00 ^e^	C: 10YR 9/2-4 S: 10YR 9/1-2
	0.25	31.28 ± 1.54 ^e^	86.97 ± 3.82 ^b^	2.19 ± 0.12 ^b^	20.43 ± 2.20 ^de^	C: 10YR 9/2-4 S: 10YR 9/1-2
	0.5	47.08 ± 3.87 ^d^	95.37 ± 18.30 ^b^	1.28 ± 0.12 ^c^	33.25 ± 10.45 ^cd^	C: 10YR 8-9/2-4 S: 10YR 9/1-2
	1	93.01 ± 7.42 ^c^	152.37 ± 24.70 ^b^	1.29 ± 0.10 ^c^	33.86 ± 2.32 ^bc^	C: 2.5Y 8.5/4 S: 2.5Y 9/2
	2	152.63 ± 6.39 ^b^	207.34 ± 6.57 ^b^	2.29 ± 0.12 ^b^	67.65 ± 4.89 ^ab^	C: 2.5Y 8.5-9/2-4 S: 2.5Y 8.5-9/2
	3	198.32 ± 4.40 ^a^	372.75 ± 111.51 ^a^	2.85 ± 0.17 ^a^	73.12 ± 4.42 ^a^	C: 2.5Y 8-8.5/4-6 S: 2.5Y 8.5-9/2-4

Note: C refers to the mushroom cap, while S refers to the mushroom stalk/stem. Different letters in the same column indicate significant differences (*p* < 0.05).

**Table 8 foods-12-03632-t008:** Vitamin D contents and color change in white shimeji and wood earmushrooms after exposure to UV irradiation for 0–3 h (cont.).

Mushrooms	UV Time (Hour)	Vitamin D2 (µg/100 g DW)	25-OH D2 (µg/100 g DW)	Ergosterol (mg/100 g DW)	Vitamin D4 (µg/100 g DW)	Color (Munsell Code)
White shimeji mushroom (*Hypsizygus tessellatus*)	0	83.73 ± 12.79 ^b^	4145.52 ± 694.53 ^ab^	96.98 ± 26.38 ^a^	196.10 ± 62.65 ^c^	C: 5Y 9/1 S: 5Y 9/1
	0.25	130.59 ± 11.42 ^b^	5941.70 ± 991.57 ^a^	100.66 ± 29.03 ^a^	370.72 ± 76.45 ^c^	C: 5Y 9/1 S: 5Y 9/1
	0.5	135.48 ± 20.20 ^b^	5081.30 ± 871.89 ^ab^	104.21 ± 1.44 ^a^	459.87 ± 45.43 ^bc^	C: 5Y 9/1 S: 5Y 9/1
	1	314.38 ± 18.42 ^a^	5192.53 ± 245.30 ^ab^	124.20 ± 12.20 ^a^	793.99 ± 86.52 ^ab^	C: 5Y 9/1 S: 5Y 9/1
	2	414.06 ± 80.25 ^a^	3797.78 ± 654.09 ^ab^	106.66 ± 25.26 ^a^	943.18 ± 223.35 ^a^	C: 5Y 9/1-2 S: 5Y 9/1-2
	3	358.77 ± 53.19 ^a^	2485.51 ± 435.16 ^b^	67.94 ± 5.53 ^a^	726.15 ± 53.69 ^ab^	C: 5Y 9/1-2 S: 5Y 9/1-2
Wood ear mushroom (*Auricularia auricula-judae*)	0	0.00 ^e^	515.14 ± 7.26 ^bc^	1.58 ± 0.33 ^b^	0.00 ^d^	C: 2.5YR 3-5/4
	0.25	114.11 ± 1.73 ^d^	707.03 ± 63.74 ^bc^	2.12 ± 0.04 ^ab^	57.81 ± 28.42 ^c^	C: 2.5YR 3-5/4
	0.5	177.21 ± 9.57 ^d^	422.94 ± 0.73 ^c^	2.13 ± 0.48 ^ab^	52.28 ± 8.61 ^c^	C: 2.5YR 3-6/4
	1	393.45 ± 39.29 ^c^	757.77 ± 58.73 ^ab^	2.29 ± 0.19 ^ab^	76.60 ± 2.83 ^c^	C: 2.5YR 3-6/4
	2	478.47 ± 7.64 ^b^	571.37 ± 101.76 ^ab^	2.04 ± 0.31 ^ab^	107.03 ± 1.71 ^b^	C: 2.5YR 2-6/4
	3	802.75 ± 23.19 ^a^	666.29 ± 92.98 ^a^	2.12 ± 0.12 ^a^	186.34 ± 13.26 ^a^	C: 2.5YR 2-6/4

Note: C refers to the mushroom cap, while S refers to the mushroom stalk/stem. Different letters in the same column indicate significant differences (*p* < 0.05).

**Table 9 foods-12-03632-t009:** Percentage of edible portion and yield factor of four cultivated mushrooms after UV irradiation (N = 3) (Mean ± SD).

Mushroom	Type of Sample	% Edible Portion	% Yield Factor	% Moisture
Bhutan oyster Mushroom (*Pleurotus sajor-caju (Fr.) Singer*)	Raw (without UV) *	95 ± 2	-	92 ± 1
	Raw *	90 ± 1	-	91 ± 0
	Boiled	88 ± 0	84 ± 4	93 ± 0
	Stir-fried	88 ± 1	69 ± 3	81 ± 1
	Grilled	86 ± 1	64 ± 2	89 ± 1
Brown hon shimeji mushroom (*Hypsizygus tessellatus*)	Raw (without UV)	95 ± 2	-	89 ± 0
	Raw	91 ± 1	-	89 ± 0
	Boiled	90 ± 1	66 ± 2	93 ± 1
	Stir-fried	91 ± 1	80 ± 3	85 ± 2
	Grilled	89 ± 3	60 ± 1	88 ± 1
Enokitake mushroom (*Flammulina velutipes*)	Raw (without UV)	87 ± 2	-	88 ± 2
	Raw	85 ± 2	-	88 ± 0
	Boiled	84 ± 2	79 ± 1	94 ± 1
	Stir-fried	85 ± 1	83 ± 4	80 ± 1
	Grilled	83 ± 1	61 ± 2	86 ± 1
Lung oyster mushroom (*Pleurotus sajor-caju (Fr.) Singer*)	Raw (without UV)	93 ± 2	-	90 ± 0
	Raw	85 ± 0	-	89 ± 0
	Boiled	84 ± 1	88 ± 7	94 ± 0
	Stir-fried	84 ± 1	72 ± 2	82 ± 2
	Grilled	85 ± 1	61 ± 5	88 ± 1

* Raw (without UV) and raw mushrooms were studied in Section 2.2 and Section 2.3 respectively.

**Table 10 foods-12-03632-t010:** Percentage of vitamin D retention of mushrooms using different cooking methods (N = 3, presented as mean ± SD).

Mushroom	Type of Cooking	True Retention of Vitamin D2 (%)	True Retention of 25-OH D2 (%)	True Retention of Vitamin D4 (%)	Weight Loss (%)
Bhutan oyster mushroom (*Pleurotus sajor-caju (Fr.) Singer*)	Boiling	88.8 ± 9.7 ^c^	66.1 ± 9.2 ^i^	80.5 ± 20.3 ^p^	16 ± 13
	Stir-frying	87.4 ± 14.6 ^c^	63.7 ± 4.6 ^i^	93.8 ± 8.0 ^p^	31 ± 3
	Grilling	87.2 ± 19.9 ^c^	100.0 ± 0.0 ^j^	74.9 ± 23.6 ^p^	36 ± 2
Brown hon shimeji mushroom (*Hypsizygus tessellatus*)	Boiling	55.5 ± 16.8 ^b^	50.2 ± 5.5 ^g^	52.1 ± 13.4 ^o^	34 ± 2
	Stir-frying	63.0 ± 32.1 ^b^	45.3 ± 12.2 ^g^	80.3 ± 17.1 ^o^	20 ± 3
	Grilling	53.4 ± 3.9 ^b^	99.8 ± 0.4 ^h^	61.8 ± 11.2 ^o^	40 ± 1
Enokitake mushroom (*Flammulina velutipes*)	Boiling	73.2 ± 6.7 ^a^	85.2 ± 25.6 ^f^	67.2 ± 6.3 ^m^	21 ± 1
	Stir-frying	89.3 ± 9.8 ^a^	84.6 ± 26.1 ^f^	88.6 ± 12.2 ^n^	17 ± 4
	Grilling	68.2 ± 13.4 ^a^	93.9 ± 8.2 ^f^	64.9 ± 8.7 ^m^	39 ± 2
Lung oyster mushroom (*Pleurotus sajor-caju (Fr.) Singer*)	Boiling	69.4 ± 2.7 ^d^	50.3 ± 12.3 ^k^	66.9 ± 6.3 ^q^	12 ± 7
	Stir-frying	75.7 ± 9.6 ^e^	46.8 ± 15.4 ^k^	80.2 ± 17.2 ^q^	28 ± 2
	Grilling	57.0 ± 8.2 ^de^	94.9 ± 8.9 ^l^	58.8 ± 6.1 ^q^	39 ± 5

Different letters in the same column indicate significant differences (*p* < 0.05).

## Data Availability

Data are contained within the article.

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
