# Peer review of "Effect of Ultraviolet Irradiation on Vitamin D in Commonly Consumed Mushrooms in Thailand"

_foods, 2023, doi:10.3390/foods12193632_

Round 1

Reviewer 1 Report

Minor corrections:

The abstract lacks a concluding statement, so I suggest the authors to include it. The table's numbering was repeated, which makes it difficult to understand. Therefore, the tables should be numbered serially and accordingly. Some tables should be converted to figures for better presentation and understanding.

Lines 35-39: Rewrite the sentence.

Lines 90-93: A figure of the UV cabinet ought to be needed here.

Line 93: Mention the make and model of the food blender and its subsequent appearance.

Line 101: Mention the color-determination device's name, make and model.

Author Response

Reviewer 1

The abstract lacks a concluding statement, so I suggest the authors to include it. The table's numbering was repeated, which makes it difficult to understand. Therefore, the tables should be numbered serially and accordingly. Some tables should be converted to figures for better presentation and understanding.

- Thank you for your valuable comments and suggestions on our manuscript.

- The conclusion was added in the revised manuscript.

- Numbering of table was serially revised.

Lines 35-39: Rewrite the sentence.

Line 35-39 in the introduction part was rewritten.

Lines 90-93: A figure of the UV cabinet ought to be needed here.

The picture of the UV cabinet was added in the Figure 1 of the revised manuscript.

Line 93: Mention the make and model of the food blender and its subsequent appearance.

The detail of food blender and other instruments were added in the revised manuscript.

Line 101: Mention the color-determination device's name, make and model.

The information of Munsell color system (Munsell®, USA) was added.

Reviewer 2 Report

The research paper titled "Effect and Stability of Ultraviolet Irradiation and Subsequent Cooking on Vitamin D Content in Commonly Consumed Mushrooms in Thailand" by Judprasong et al. provided valuable insights into the importance of consuming foods that are rich in Vitamin D. It was an informative and very useful study.

Additional comments:

It is crucial to mention the optimal time for maximizing vitamin D production through UV-B irradiation when writing the abstract. It is recommended to include a figure of the mushroom's physical characteristics in section 3.21. The figure is provided in the supplemental section.

To ensure uniformity, present all units as either µg/100g or µg per 100g. It is also advisable to include UHPLC-MS-MS chromatograms of vitamin D components for identification purposes.

Lines 107-111: In this section, it is important to clearly state the volumes of extraction solvent used.

 Line 114: Correct the column diameter 150 mm

 can be improved

Author Response

Reviewer 2

The research paper titled "Effect and Stability of Ultraviolet Irradiation and Subsequent Cooking on Vitamin D Content in Commonly Consumed Mushrooms in Thailand" by Judprasong et al. provided valuable insights into the importance of consuming foods that are rich in Vitamin D. It was an informative and very useful study.

Additional comments:

It is crucial to mention the optimal time for maximizing vitamin D production through UV-B irradiation when writing the abstract. It is recommended to include a figure of the mushroom's physical characteristics in section 3.21. The figure is provided in the supplemental section.

- Thank you for your valuable comments and suggestions on our manuscript.

- Figure and physical characteristics of the mushroom was added in Table 3-4 in the revised manuscript.

To ensure uniformity, present all units as either µg/100g or µg per 100g. It is also advisable to include UHPLC-MS-MS chromatograms of vitamin D components for identification purposes.

The UHPLC-MS-MS chromatograms of vitamin D components was added in Figure 2.

Lines 107-111: In this section, it is important to clearly state the volumes of extraction solvent used.

The volume of all solvents were added.

Line 114: Correct the column diameter 150 mm

The correct diameter of column was revised.

Ensure uniformity, present all units as either µg/100g or µg per 100g

The unit of µg/100g was used throughout the revised manuscript.

Lines 107-111: In this section, it is important to clearly state the volumes of extraction solvent used.

The volume of all solvents were added.

 Line 114: Correct the column diameter 150 mm

The diameter of column was revised.

Reviewer 3 Report

The methods revealed that mushroom samples were divided into three different sizes, but this parameter was not investigated in the study. What was the effect of size on the exposure time of UV-B? There could be a potential bias here and must be mentioned.

There are some data that must be presented in the manuscript. In particular, there four different times of boiling were tested to assess vitamin D in four types of mushrooms, a total of 12 different cooking times from 3 different cooking methods. Yet, Table 5 only shows incomplete data sets, noting also only 3 vitamers were reported here.

It would be suggested that authors state the hypotheses, whether the data support or contradict them can be discussed and explain why or why not, and discuss possible reasons for unexpected results.

Importantly, address any limitations in the data, such as sample size, data collection methods, or potential biases. This shows a well-rounded interpretation that considers potential shortcomings.

Abstract

The content is quite clear, though this can be improved by mentioning which D vitamers were investigated.

L 21 according to ‘time of irradiation’ could be replaced with another wording to improve understanding.

L 22 in which mushrooms?

L 23 true retention in cooked mushrooms?

Also, please correct the grammatical errors.

Introduction

Punctuation and grammatical errors must be addressed. Some sentences are truncated and must be corrected.

L 46-52 re-write this paragraph for clarity.

The intro is well written, providing sufficient information to set the scene of the study. This can be improved by making it more cohesive.

Methods

Please clarify the sample treatments to improve understanding, where each type of mushroom was categorised by sizes (L, M and S), please confirm that there were 24 groups of samples prepared before UV-B. Each group was then treated with different exposure times. Were there two variables investigated here? A schematic diagram would be helpful to visualise what the experiments are about. As they continued with the investigation of four types of mushrooms. How many grams of mushrooms were placed in the cabinet?

L 93 unclear which samples are referred to here, e.g. UV-B treated samples?

Food blender or food processor? Please include the name of the maker?

L 93 ‘were lay’ should be replaced with different wording. Re-phrase the sentence for clarity and keep it concise.

L 95 homogenised samples?

L 98 ground with which tool? Into powder?

L insert concentration of IS

L 106 validated method? Please confirm.

L 107-112 Was this described in detail in the previous publication? E.g molarity of solutions or solvents used?

L 111 which pore size was the syringe filter

L 114 Check the column name

L 115 Which vitamers were detected and quantified? What was the concentration range used for quantification?

Kindly provide a chromatogram of real samples in the supplementary.

L 127 what was the optimum condition?

Boiling of samples: Was the water used for cooking also analysed to determine whether there could be any D vitamers leaching out into the water? Explain and justify.

Stir frying: Was cooking oil was used for stir-frying the mushrooms? What type of oil was used? Was the oil also analysed to determine any possible leaching of vitamin D? Explain and justify.

Table 2. explain and justify the different cooking time used for each mushroom type and the impact on vit D contents.

Explain and justify the storage at minus 20 deg in terms of degradation loss due to the sensitivity of the vitamers. How long was the storage prior to analyses?

L 147 Clarify that sample weights were recorded by using only 2 significant digits.

How many data points were collected from the analyses of D vitamers? How many sample replicates were prepared?

Results and discussion

3.1. Which mushrooms of the same species? This has not been established earlier, hence this section is not clear at all.

3.2. Acronyms used in the manuscript must be defined when first appear.

L 173 Measured D2 is based on wwb or dwb?

What is the LOQ of D2? Explain and justify, whether it was due to the sensitivity of the method or was it really due to the absence of D2 in samples?

L 181 what does it mean ‘followed by rime exposure’?

L 184-186 the analysed D2 must be higher than LOQ in order to support the claim of the significant increase.

Table 3. Analysed values presented in the table should be based on dwb to accurately compare the levels of vitamers between treatments.

3.2.2 This paragraph is not cohesively written and data are not discussed within context. Write clearly when comparing data between samples, in the current state the content is confusing.

3.2.3. This section must be rewritten to avoid misunderstanding, which requires some interpreted data that must be written correctly. It might be better to express the increase in vitamers as a percentage, if possible, for example, L 243. As mentioned earlier, all concentrations must be expressed as dwb. Results are not well discussed, too speculative and out of the scope of the study given that consumption and dietary intakes are not the focus of the study.

3.2.4. L 253-254 please re-write for clarity. Data must be expressed in dwb, allowing comparison with published data. L 265-268 this sentence is disjointed, please re-write.

3.2.5 re-write the section for clarity. Write concisely and correct the grammatical errors.

3.3.1. this section is very subjective, removing in-edible parts was dependent on personal intervention. Explain and justify the significance of this section.

3.3.2. L 291 delete the stability of vit D2 from this section for clarity and keep the section brief with no repetitive statement.

Overall, sections 3.3.1, 1.1.2 and 3.3.3 must be kept brief as these are not the focus of the study.

L 3.3.4 This section is worth more discussion rather than only presenting the results. As mentioned previously in the methodology, leaching and other factors might contribute to low retention shown in the table which could be discussed further.

L 329-333 repetitive statement and must be deleted. Refer to the previous section already describing this.

L 339 competent is not the correct choice of word and must be replaced.

Table 5: Please declare the cooking times on the data presented here, as per Table 2. 

Writing issues: Write concisely throughout the manuscript, for example, establish at the beginning that the types of mushrooms selected for the study were cultivated ones, hence, only ‘mushrooms’ can be written, and this clearly corresponds to the term established earlier.

Author Response

Reviewer 3

The methods revealed that mushroom samples were divided into three different sizes, but this parameter was not investigated in the study. What was the effect of size on the exposure time of UV-B? There could be a potential bias here and must be mentioned.

Thank you for your valuable comments and suggestions on our manuscript. For representative of the harvest mushroom in each batch, each mushroom into three different sizes.  This use for comparison of each UVB irradiation time.  We did not investigate the effect of size in this study.

There are some data that must be presented in the manuscript. In particular, there four different times of boiling were tested to assess vitamin D in four types of mushrooms, a total of 12 different cooking times from 3 different cooking methods. Yet, Table 5 only shows incomplete data sets, noting also only 3 vitamers were reported here.

We agreed with reviewer’s comments. However, only four types of representative mushrooms with high amount of vitamin D were chosen in this section.  The 3 different cooking methods were used based on the commonly consumed pattern in Thai’s mushrooms. Table 10 (revised) used only vitamin D2 for study of true retention, this is due to the beneficial to the health is only vitamin D2 and D3.  However, vitamin D3 is not found in the mushrooms.

It would be suggested that authors state the hypotheses, whether the data support or contradict them can be discussed and explain why or why not, and discuss possible reasons for unexpected results.

The hypothesis of this study is UVB irradiation can increase the level of vitamin D in mushrooms.  The results in this study supported the hypothesis. The discussion was also mentioned in the introduction part and section 3.2 Effect of UV irradiation on vitamin D contents in mushrooms.

Importantly, address any limitations in the data, such as sample size, data collection methods, or potential biases. This shows a well-rounded interpretation that considers potential shortcomings.

The limitation was also added in the conclusion.

Abstract:

The content is quite clear, though this can be improved by mentioning which D vitamers were investigated.

L 21 according to ‘time of irradiation’ could be replaced with another wording to improve understanding.

L 22 in which mushrooms?

L 23 true retention in cooked mushrooms?

Also, please correct the grammatical errors.

- The word “exposure time” was used instead of “time of irradiation”.

- The name of mushroom “enokitake mushrooms” was added in the abstract.

- The sentence was rewritten.

Introduction:

Punctuation and grammatical errors must be addressed. Some sentences are truncated and must be corrected.

L 46-52 re-write this paragraph for clarity.

The intro is well written, providing sufficient information to set the scene of the study. This can be improved by making it more cohesive.

The information was added in the revised manuscript.

Methods

Please clarify the sample treatments to improve understanding, where each type of mushroom was categorised by sizes (L, M and S), please confirm that there were 24 groups of samples prepared before UV-B. Each group was then treated with different exposure times. Were there two variables investigated here? A schematic diagram would be helpful to visualise what the experiments are about. As they continued with the investigation of four types of mushrooms. How many grams of mushrooms were placed in the cabinet?

- L 93 unclear which samples are referred to here, e.g. UV-B treated samples? Food blender or food processor? Please include the name of the maker?

- L 93 ‘were lay’ should be replaced with different wording. Re-phrase the sentence for clarity and keep it concise.

- L 95 homogenised samples?

- L 98 ground with which tool? Into powder?

- L113 insert concentration of IS

- L 106 validated method? Please confirm.

- L 107-112 Was this described in detail in the previous publication? E.g molarity of solutions or solvents used?  

- L 111 which pore size was the syringe filter

- L 114 Check the column name  

- L 115 Which vitamers were detected and quantified? What was the concentration range used for quantification?

- Kindly provide a chromatogram of real samples in the supplementary.

- L 127 what was the optimum condition?

Boiling of samples: Was the water used for cooking also analysed to determine whether there could be any D vitamers leaching out into the water? Explain and justify.

- Stir frying: Was cooking oil was used for stir-frying the mushrooms? What type of oil was used? Was the oil also analysed to determine any possible leaching of vitamin D? Explain and justify.

- Table 2. explain and justify the different cooking time used for each mushroom type and the impact on vit D contents.

- Explain and justify the storage at minus 20 deg in terms of degradation loss due to the sensitivity of the vitamers. How long was the storage prior to analyses?

- L 147 Clarify that sample weights were recorded by using only 2 significant digits.

- How many data points were collected from the analyses of D vitamers?

- How many sample replicates were prepared?

- The samples were divided into six parts, each part containing large, medium, and small sizes ( not 24 groups).

- Analysis of Variance (two-way ANOVA) and Duncan’s Multiple Range Test were used.

- There are 10 Tables and 2 Figures, therefore more explanation was added in this part. “These samples (2 kg) were then divided into six groups of the same amount (about 300 g/group) and each group containing large, medium, and small sizes before being placed into a UV cabinet to be exposed to UV-B irradiation for 0, 0.25, 0.5, 1, 2, and 3 hours (six study groups).”

- This sentence (L93) was revised as “The UV-B treated samples were homogenized in a food blender (MR-1268, MARA®, Thailand).”

- L93, the word “lay on” was replaced by “put in”.

- L95, the word “homogenized samples” was replace by “homogenized UV-B treated samples”.

- L98, the brand name and country were added.

- L113, concentration was added as “Four ppm of trideuterated 2H3D2 (50 µL) was used as an internal standard.”

- L 106, the full validated method was published in ref 29 (revised manuscript).

- L 107-112, The detail of solution and solvents used were added.

- L111, A 0.2 µm. NYLON syringe filter was added.

- L114, the column name (Acquity uplc, Waters, Ireland) was added.

- L115, Six vitamers of vitamin D (Vitamin D2, Vitamin D3, Ergosterol, 7-DHC, 25-OH D2, 25-OH D3, and Vitamin D4) and 2H3D2 (internal standard) were separated and detected by UHPLC-MS-MS).

- UHPLC-MS-MS chromatograms of vitamin D components is added in Figure 2.

- The optimum condition was previously study and published in ref 29. The condition used is presented in Table 2. However, the vitamin D did not analyzed in the cooking water. We studied only in the cooked mushroom (commonly consumed pattern in Thailand).

- The information of Stir-frying with small amount of palm oil was added. This sample included he palm oil used and small water leaching from the mushrooms.

- The cooking time used are depended on the type of mushrooms which we used the same time of cooking for each mushroom. The small impact was included in the range of true retention of vitamin D.

- Sample stored at -20oC is used for moisture analysis.  For vitamin D, samples were dried by freeze dryer, ground into fine powder, and then stored at -20oC.  They were analyzed vitamin D within 1 month which these condition could preserve the vitamin.

- L 147, sample weight used 4 digits analytical balance. Whereas for weight before and after cooking used at least 2 digit balance depending on the maximum weight of each balance.

- For each vitamers of vitamin D, the MS-MS set up at 7 data points of each peak of vitamers.

- Each analysis was usually done in duplicate.

Results and discussion

- 3.1. Which mushrooms of the same species? This has not been established earlier, hence this section is not clear at all.

- 3.2. Acronyms used in the manuscript must be defined when first appear.

- L 173 Measured D2 is based on wwb or dwb?

What is the LOQ of D2? Explain and justify, whether it was due to the sensitivity of the method or was it really due to the absence of D2 in samples?

- L 181 what does it mean ‘followed by rime exposure’?

- L 184-186 the analysed D2 must be higher than LOQ in order to support the claim of the significant increase.

- Table 3. Analysed values presented in the table should be based on dwb to accurately compare the levels of vitamers between treatments.

- 3.2.2 This paragraph is not cohesively written and data are not discussed within context. Write clearly when comparing data between samples, in the current state the content is confusing.

- 3.2.3. This section must be rewritten to avoid misunderstanding, which requires some interpreted data that must be written correctly. It might be better to express the increase in vitamers as a percentage, if possible, for example, L 243. As mentioned earlier, all concentrations must be expressed as dwb. Results are not well discussed, too speculative and out of the scope of the study given that consumption and dietary intakes are not the focus of the study.

- 3.2.4. L 253-254 please re-write for clarity. Data must be expressed in dwb, allowing comparison with published data. L 265-268 this sentence is disjointed, please re-write.

- 3.2.5 re-write the section for clarity. Write concisely and correct the grammatical errors.

- 3.3.1. this section is very subjective, removing in-edible parts was dependent on personal intervention. Explain and justify the significance of this section.

- 3.3.2. L 291 delete the stability of vit D2 from this section for clarity and keep the section brief with no repetitive statement.

- Section 3.1 was revised.

- The full name of abbreviation was added.

- L173, The vitamin D results in Table 5-8 (revised manuscript) was changed from fresh weight to dried weight. All the data of vitamers are higher than the LOQ which published in ref 29.

- L 181, typing error of “rime” was revised to “time”.

- All the data of vitamers are higher than the LOQ which published in ref 29.

- As suggested, The vitamin D results in Table 5-8 (revised manuscript) was changed from fresh weight to dried weight.

- The result was modified and presented as dried weight and fresh weight.

- The percentage of changed was presented as suggestion.

- There are a limitation of study in this vitamer in mushroom, the discussion of ref 26 and 27 were presented.

- The percentage of changed was presented as suggestion.

- There are a limitation of study in this vitamer in mushroom, the discussion of ref 26 and 27 were presented.

- This section is explained the correlation of vitamin D2 and vitamin D4, may we did not change the sentence.

- The edible part is benefit for the consumer when buy a food (e.g. mushrooms in this case), only some part can be eaten.  In addition, only the edible part was used in this study.

- The stability was removed from this section.

- Overall, sections 3.3.1, 3.1.2 and 3.3.3 must be kept brief as these are not the focus of the study.

L 3.3.4 This section is worth more discussion rather than only presenting the results. As mentioned previously in the methodology, leaching and other factors might contribute to low retention shown in the table which could be discussed further. L 329-333 repetitive statement and must be deleted. Refer to the previous section already describing this.

- In term of food composition database and nutrition, the results in these sections are very important.  However, there are limitation of data comparison, may we kept in sections for other user group.

- L 339 competent is not the correct choice of word and must be replaced.

- The word “competent” was changed to “interesting”.

- Table 5: Please declare the cooking times on the data presented here, as per Table 2. 

The cooking time of each mushroom is done and presented in Table 2.

Round 2

Reviewer 3 Report

Thank you for addressing the comments provided to authors, the manuscript reads better and is much easier to understand.

It is though noticed that in some cases, the extra information provided in the rebuttal has not been included in the manuscript. E.g samples were prepared in duplicate. If this were done, the contents of the manuscript would be better.

Introduction

L 56 Promote the health benefits of vit D

Methods

Thank you for describing section 2.1 with clarity: different types of mushrooms and different exposure times. Time = 0 served as control.

L 122 vit D method seems not to be searchable in the public domain, please explain and correct the citation/reference.

Section 3.2.1 thank you for making this paragraph much clearer.

L 258 were not significantly different, so correct the p value. Please check this claim through the whole manuscript and correct it accordingly. E.g L 376.

There are grammatical errors still found in the manuscript and they must be corrected.

Correct the grammatical error of the title: Effect of Ultraviolet Irradiation and Subsequent Cooking on the Stability Vitamin D Content in Commonly Consumed Mushrooms in Thailand

Author Response

Comments and Suggestions for Authors

Thank you for addressing the comments provided to authors, the manuscript reads better and is much easier to understand.

Thank you for your valuable comments and suggestions on our manuscript. We appreciate the time and effort you have dedicated to reviewing our work. 

It is though noticed that in some cases, the extra information provided in the rebuttal has not been included in the manuscript. E.g samples were prepared in duplicate. If this were done, the contents of the manuscript would be better.

Vitamin D and moisture content in this study were analyzed in duplicate.  As suggested, this is mentioned in session 2.2.2 Vitamin D determination and session 2.3.4 Moisture determination

Introduction

L 56 Promote the health benefits of vit D

Corrected this sentence. Line 58 Change to promote the health benefits of vitamin D

Methods

Thank you for describing section 2.1 with clarity: different types of mushrooms and different exposure times. Time = 0 served as control.

Thank you very much.

L 122 vit D method seems not to be searchable in the public domain, please explain and correct the citation/reference.

Reference of Association of Official Analytical Chemists (AOAC) 2019 was added in the reference No. 21.

Section 3.2.1 thank you for making this paragraph much clearer.

Thank you very much.

L 258 were not significantly different, so correct the p value. Please check this claim through the whole manuscript and correct it accordingly. E.g L 376.

The significant different was checked and the following sentence was modified as “The content of 25-OH D2 in both control (unexposed) and treated samples (ex-posed) was found significantly (p < 0.05) highest in white shimeji mushrooms, followed by wood ear, straw, and enokitake mushrooms, respectively (Table 5-8).”.

Correct the grammatical error of the title: Effect of Ultraviolet Irradiation and Subsequent Cooking on the Stability Vitamin D Content in Commonly Consumed Mushrooms in Thailand

The title was changed to “Effect of Ultraviolet Irradiation on the Vitamin D in Commonly Consumed Mushrooms in Thailand”.

Comments on the Quality of English Language

There are grammatical errors still found in the manuscript and they must be corrected.

The gramma was checked throughout the revised manuscript.
